# Belief in sexism shift: Defining a new form of contemporary sexism and introducing the belief in sexism shift scale (BSS scale)

**Miriam K. Zehnter**[1]*, **Francesca Manzi**[2], **Patrick E. Shrout**[3], **Madeline E. Heilman**[3]

**1** Department of Work, Economic, and Social Psychology, University of Vienna, Vienna, Austria, **2** Department of Social, Health, and Organisational Psychology, Utrecht University, Utrecht, The Netherlands, **3** Department of Psychology, New York University, New York, New York, United States of America

\* miriam.zehnter@univie.ac.at

## Abstract

The belief that the target of sexism has shifted from women to men is gaining popularity. Yet despite its potential theoretical and practical importance, the belief that men are now the primary target of sexism has not been systematically defined nor has it been reliably measured. In this paper, we define the *belief in sexism shift* (BSS) and introduce a scale to measure it. We contend that BSS constitutes a new form of contemporary sexism characterized by the perception that anti-male discrimination is pervasive, that it now exceeds anti-female discrimination, and that it is caused by women's societal advancement. In four studies (N = 666), we develop and test a concise, one-dimensional, 15-item measure of BSS: the BSS scale. Our findings demonstrate that BSS is related to, yet distinct from other forms of sexism (traditional, modern, and ambivalent sexism). Moreover, our results show that the BSS scale is a stable and reliable measure of BSS across different samples, time, and participant gender. The BSS scale is also less susceptible to social desirability concerns than other sexism measures. In sum, the BSS scale can be a valuable tool to help understand a new and potentially growing type of sexism that may hinder women in unprecedented ways.

## Introduction

> *"Man has been the dominant sex since, well, the dawn of mankind. But for the first time in human history, that is changing–and with shocking speed."* [1].

In 2016, two former Yahoo employees filed independent lawsuits alleging that the company had discriminated against them because of their gender [2]. These were not isolated cases. Several large corporations and educational institutions (e.g., Google, Yale University) were recently confronted with similar claims of gender discrimination [3,4]. These allegations were surprising not because of the nature of the complaint–thousands of gender-based discrimination suits are filed each year in the US alone [5]–but because of the gender of the complainants: All of the plaintiffs were men. Notably, many of these lawsuits were filed in domains that remain heavily male-dominated.

**Data Availability Statement:** All data, analyses, and R code are available on the Open Science Framework via https://www.doi.org/10.17605/OSF.IO/2DGTE

**Funding:** This research was funded by the Marietta Blau-Stipendium of the Austrian Agency for International Cooperation in Education and Research (OeAD) under the Austrian Federal Ministry of Education, Science, and Research (Project number: ICM-2015-03097), and by the Förderstipendium of the University of Vienna. The publication of this manuscript was funded through the Open Access Publication Fund of the University of Vienna. The funding agencies had no role in study design, data collection and analysis, decision to publish, or preparation of the manuscript.

**Competing interests:** The authors have declared that no competing interests exist.

The belief that men are now the disadvantaged gender has gained traction, sparking debates both in popular culture and academia [6–8]. Data indicate that men, in particular, believe that they are more likely than ever before to suffer gender-based discrimination [9,10]. About two thirds of men claim to have faced at least some discrimination due to their gender [11]. Even some women share these beliefs. In the United States, 9% women (and 15% men) believe that women have better job opportunities than men [12], and 5% of women (and 14% of men) think that it is now easier to be a woman than a man [13].

Taken together, these data suggest that there is an increasing number of men and women who believe that "the tables have turned"–that is, that the target of sexism has shifted from women to men, and that anti-male discrimination is now a pervasive social issue. Recent efforts have been made to begin to understand this emerging belief as well as its potential causes and consequences [14–16]. However, the literature to date has not yet offered a systematic definition of what this belief entails, nor have researchers provided a reliable way to measure it.

The aim of this paper is to introduce and formally define the *belief in sexism shift* (BSS) and to develop a scale to measure it: the BSS scale. We first provide a definition of BSS and place it in the broader context of contemporary anti-female sexism. We then present research on the development and validation of the BSS scale, a 15-item self-report measure reflecting the perceptions of anti-male discrimination that comprise BSS. Finally, we discuss how the BSS scale may help advance our understanding of sexism, in general, and enable future research on BSS, in particular.

## Defining belief in sexism shift

The belief in sexism shift centers around the victimization of men. It is characterized by the belief that men have become the primary victims of sexism, that male victimization is pervasive, and that this new form of discrimination is the result of women's societal advancement.

As its name suggests, those who endorse BSS perceive a *shift* or transition from anti-female sexism to anti-male sexism. This shift is thought to have occurred recently and to have increased to a point where men now suffer more discrimination than women [9,10]. Men are not merely seen as additional victims of rigid gender norms alongside women; but rather, as the *primary* target of gender discrimination.

Importantly, male victimization is not limited to contexts where men may actually be discriminated against, such as female-dominated professions or traditionally feminine roles [17]. Rather, individuals who endorse BSS perceive anti-male discrimination to be ubiquitous, manifesting in multiple settings (e.g., the workplace, politics), through multiple perpetrators (e.g., the media, feminists), and in multiple ways (e.g., political correctness, devaluation of masculinity; [18–20]).

In addition, BSS entails that it is women's societal advancement that has, to a great extent, led to anti-male discrimination. Instead of viewing women's progress as a step towards greater gender equality for both women and men, those endorsing BSS assume that women's gains entail losses for men [9,14,21]. This zero-sum perspective on gender discrimination is important: it differentiates individuals who endorse BSS from those who perceive that all genders may suffer from some form of discrimination [22–24].

## Belief in sexism shift as a contemporary form of anti-female sexism

On the surface, the focus of BSS is on men and not on women. Thus, the idea that BSS constitutes a new form of anti-female sexism may seem counterintuitive. We argue, however, that BSS is consistent with both the definition and function of anti-female sexism.

Sexism has been defined as discriminatory and prejudicial attitudes, beliefs and practices directed against a person based on their sex and/or gender [25]. In the case of BSS, negative attitudes towards women are concealed through a narrative of male victimization where men's

disadvantages are the product of a system that has relentlessly favored women. By implying that women's progress has been attained through favoritism, BSS subtly discounts women's abilities and downplays their merits in their own advancement.

Further, like other forms of anti-female sexism, the function of BSS is to sustain a gender hierarchy that places men over women [26]. However, BSS fulfills this function in new and specific ways. Namely, BSS diverts attention from ongoing discrimination against women by redirecting the focus, placing men as the targets of an ostensibly anti-male system. Not only does BSS obscure anti-female discrimination, but it also goes a step further, positing that women now have an unjust advantage over men. This implies that any measure to further advance women is unnecessary and effectively hurts men.

In sum, BSS serves to uphold men's higher social status by providing an unprecedented rationale for prioritizing men's rights over women's rights. From the perspective of those who endorse BSS, efforts to attain gender equality should effectively move away from women's issues to instead focus solely on the protection of men's rights.

## Differences between belief in sexism shift and other forms sexism

BSS, although a manifestation of sexism against women, differs from the most well-studied forms of anti-female sexism in critical ways.

**Traditional sexism.**    Traditional or "old-fashioned" sexism focuses on the social roles that women and men should fulfill [27,28]: While intellectual and leadership roles are ascribed to men, housekeeping and caregiving roles are ascribed to women. Thus, traditional sexism serves to justify the unequal treatment of women and men by overtly upholding conventional gender roles [27,28]. Unlike traditional sexism, upholding BSS does not require a traditional view of gender. Instead, the outward focus of BSS is on gender equality–albeit to advance men. In this way, BSS provides a more modern and socially acceptable narrative to justify placing men over women and is thus a much more subtle form of sexism than traditional sexism.

**Ambivalent sexism.**    Ambivalent sexism assigns both negative (hostile sexism) and positive (benevolent sexism) attributes to women. While blatant hostility is directed towards non-traditional and feminist women, traditional women are seen as needing and worthy of protection [29]. BSS does not explicitly address any particular subgroup of women–all women are seen as benefitting from anti-male discrimination. Like hostile sexists, those who endorse BSS hold negative views of women. However, BSS is much less blatant: it conceals negative attitudes towards women behind a narrative of male victimhood. Thus, unlike hostile sexism, BSS may not be readily recognized as anti-female, making it a more insidious form of contemporary sexism. In addition, although both benevolent sexism and BSS are subtle forms of sexism, they each obscure anti-female attitudes in markedly different ways. While the former uses paternalism and praise, the latter achieves this goal by shifting the focus towards the victimization of men.

**Modern sexism.**    Rather than focusing on women's roles or attributes, modern sexism centers around the current state of gender discrimination. Indeed, modern sexism is defined by the denial of structural discrimination against women [28]. Although modern sexists do not explicitly belittle women, denying anti-female discrimination allows them to subtly trace back the causes for women's social stagnation to women's own shortcomings, rendering further measures for the advancement of women obsolete. BSS, also, focuses on the state of gender discrimination. But unlike modern sexists, those endorsing BSS posit that gender discrimination is an ongoing societal issue, albeit with different victims. Moreover, if the current victims of discrimination are now men, measures to ensure women's advancement are not only seen as obsolete, but also as a barrier to attaining true gender equality. Thus, those who uphold BSS see measures to promote men (over women) as both legitimate and necessary.

## Overview of the present research

The goal of this research is to provide a reliable self-report measure of the tendency to endorse the belief in sexism shift and to build a case for its validity. This is a first and essential step for future research on BSS, its psychological underpinnings, and its downstream consequences. In the following sections, we present four studies designed to develop and test a concise, one-dimensional, 15-item measure–the BSS scale. In a pilot study, we use principal component analysis (PCA) to assess the plausibility of a one-dimensional structure of BSS in an initial pool of 75 items. We also begin to reduce the item pool. In Study 1, we use exploratory factor analysis (EFA) to identify the items that are related to the core construct. After selecting 15 items that balance conceptual relevance and psychometric credentials, we use confirmatory factor analysis (CFA) for a preliminary examination of the fit of a one-dimensional psychometric model. In Study 2, we use CFA to assess the factor structure of the BSS scale with a new sample. Then, we assess the BSS scale's convergent and discriminant validity. In addition, we examine the test-retest reliability of the BSS scale over a time span of two weeks. Finally, in Study 3, we perform analyses of measurement invariance (AMI) to assess whether the BSS items have the same psychometric structure among women and men. An overview of the four studies' aims and statistical analyses is provided in Table 1.

# General method

Before presenting our results, we will identify methodological aspects that were consistent across all studies.

## Research participants

Six hundred and sixty-six adults were involved in the studies reported here. Of these participants, 288 (43%) identified as women, 376 (56%) identified as men, and two (<1%) identified as non-binary. All participants were recruited via Amazon's Mechanical Turk (MTurk) and paid between 0.75 and 2.00 US Dollars. Given the language and wording of some of the scale

**Table 1. Overview of the four studies.**

|  | **Study aims** | **Statistical analyses** |
|---|---|---|
| Pilot Study | Test one-dimensional structure of BSS and reduce the initial list of items | Principal component analysis (PCA) |
| Study 1 | Select items for 15-items BSS-scale | Exploratory factor analysis (EFA) |
|  | Preliminary test of factor structure of the BSS-scale | Confirmatory factor analysis (CFA) |
| Study 2 | Test of factor structure of the BSS-scale | CFA |
|  | Test of convergent validity of the BSS-scale | Pearson product-moment correlations of the BSS-scale with established sexism scales (i.e., traditional, modern, hostile, and benevolent sexism) |
|  | Test of discriminant validity of the BSS-scale | Confirmatory factor analyses to compare unitary one-factor vs. discriminant two-factor models including the BSS-scale and one other sexism scale (i.e., traditional, modern, hostile, and benevolent sexism) |
|  |  | Comparisons of Pearson product-moment correlations of the BSS-scale and other sexism scales with social desirability |
|  | Test-retest reliability of the BSS-scale | Pearson product-moment correlation of BSS at time 1 with BSS at time 2 |
| Study 3 | Test whether the BSS-scale measures the same construct among women and men | Analysis of measurement invariance |

items (e.g., "In the US, discrimination against men is on the rise"), participants were also required to reside in the United States and to be at least 18 years of age.

**Identification of careless responses.** Although MTurk generally produces high-quality data [30], we took additional steps to identify careless responses. Following recommendations [31] we included one instructed-response item per 50 to 100 items (e.g., "Reading the items carefully is critical, if you are paying attention please choose '*I strongly agree*'"). We also computed a psychometric antonym index by correlating item pairs with opposite content (e.g., "Feminism is about favoring women over men" and "Feminism does not discriminate against men"). Since such items are expected to correlate negatively, positive indices indicate careless response [31]. In addition, we screened the responses of participants who completed the questionnaire suspiciously fast, that is, those with an average response time of under two seconds per item [32]. Individuals were excluded from data analysis either for incorrect response(s) to the instructed response item(s), or for rushed response times in combination with a psychometric antonym index above .30.

Table 2 provides a summary of the sample characteristics, including socio-demographic information, by study.

## Procedure

The research presented here was approved by New York University's Internal Review Board (NYU IRB). All studies were conducted on MTurk and introduced as a study on "people's opinions about current issues". Upon providing informed consent, participants were asked to complete one or more questionnaires and to provide basic demographic information (gender, race/ethnicity, and age). The order of the items and questionnaires (when applicable) was randomized in every study. After completing the study, participants were fully debriefed.

**Table 2. Sample characteristics by study.**

|  | Pilot Study N (%) | Study 1 N (%) | Study 2 N (%) | Study 2 (retest) N (%) | Study 3 N (%) |
|---|---|---|---|---|---|
| **Total amount of responses** | 119 (100) | 239 (100) | 378 (100) | 217[1] (100) | 736[2] (100) |
| **Excluded for careless responses** | 7 (6) | 20 (8) | 43 (11) | 11 (5) | 70[3] (10) |
| **Final sample size** | 112 (94) | 219 (92) | 335 (89) | 206 (95) | 664 (90) |
| **Gender** |  |  |  |  |  |
| Women | 48 (43) | 95 (43) | 145 (43) | 89 (43) | 288 (43) |
| Men | 63 (56) | 123 (56) | 190 (57) | 116 (57) | 376 (57) |
| Not binary | 1 (< 1) | 1 (<1) |  |  | 2 (< 1) |
| **Race** |  |  |  |  |  |
| Asian/ Asian American | 4 (4) | 19 (9) | 28 (8) | 16 (8) | 51 (8) |
| Black/ African American | 16 (14) | 27 (12) | 17 (5) | 9 (4) | 60 (9) |
| Hispanic/ Latin American | 7 (6) | 15 (7) | 26 (8) | 15 (7) | 48 (7) |
| Mixed Races | 1 (< 1) | 7 (4) | 9 (3) | 6 (3) | 17 (3) |
| Native American |  | 1 (< 1) | 2 (< 1) | 1 (< 1) | 3 (< 1) |
| White/ European American | 84 (75) | 150 (68) | 249 (74) | 155 (76) | 483 (73) |
| Not specified |  |  | 4 (< 1) | 3 (< 1) | 4 (< 1) |
| **Mean age (SD)** | 33.82 (8.58) | 35.06 (10.37) | 35.60 (10.71) | 36.14 (10.37) | 35.17 (10.26) |
| **Age range in years** | 18–58 | 18–67 | 19–71 | 19–71 | 18–71 |

[1] Note. The retest response rate was 65 percent.

[2] Note. This sample was obtained by collapsing responses from the Pilot Study, Study 1, and Study 2.

[3] Note. Two additional participants indicating non-binary gender were excluded from analysis of measurement invariance.

### Item generation

We used various approaches to develop items for the BSS scale. First, we invited a group of experts on gender and intergroup relations to brainstorm items that could reflect potential manifestations of BSS. Our second approach involved the examination of relevant websites (e.g., avoiceformen.com), online forums (e.g., reddit), and recent media articles and debates [33,34] for "real-world" examples of BSS. In a third and final approach, we turned to the literature on sexism and related constructs. Several items were inspired by recent work on current perceptions of gender discrimination [10]. We also rephrased items from previous sexism scales, such as the Modern Sexism Scale [28] and the Ambivalent Sexism Inventory [29], to reflect BSS. For example, the modern sexism item "Women often miss out on good jobs due to sexual discrimination" was adapted to read "Men often miss out on good jobs due to affirmative action for women".

This process resulted in an initial list of 75 items reflecting male victimization, the central characteristic of BSS. The items were created to describe beliefs about anti-male discrimination on a general level (e.g., "In the US, discrimination against men is on the rise") as well as specific examples of its multiple manifestations, perpetrators, and expressions (e.g., "Nowadays, men don't have the same chances in the job market as women"). They also included statements reflecting zero-sum beliefs about gender discrimination (e.g., "Giving women more rights often requires taking away men's rights"). All items were worded as statements, and responses were coded using a 7-point Likert scale (1 = *I strongly disagree*, 7 = *I strongly agree*). 16 items were reverse-coded to provide within-scale attention checks.

## Pilot

The goal of the pilot study was to assess the plausibility of a one-dimensional structure of BSS in the initial item pool of 75 items. We also sought to select a subset of items that were strongly related to the dominant dimension.

### Participants and procedure

We analyzed the responses of one hundred and twelve participants (for detailed sample characteristics, see Table 2). The sample size was chosen to provide an initial check on how highly correlated the item-responses were. With this sample size, the 95% confidence bound on a sample correlation of zero was ±.19 (and narrower for positive correlations), giving us the ability to identify the items that were clearly uncorrelated with the first principle component of the item set. The structure of the selected items was verified in subsequent studies with larger sample sizes.

### Data analysis

Because the sample size was relatively small, we analyzed the item correlation matrix using principal components analysis (PCA) rather than factor analysis. The pattern of eigenvalues from PCA reveals the extent to which the items reflect overlapping variance. Corresponding to the first eigenvalue is an eigenvector, which gives an approximation to factor loadings from a one-factor model. We computed the eigenvalues of the 75 by 75 item matrix and determined the number of components by examining the scree plot of the eigenvalues and using Horn's parallel analysis [35]. Then, we performed PCA on a one-component and a two-component solution. Adhering to a theory-driven approach, we checked to see if items that are key markers of the BSS construct had high loadings on the first eigenvector. Specifically, we examined whether the items reflecting male victimization on a general level loaded highly on the first

component. We then selected a subset of items that balanced conceptual relevance and psycho-metric credentials. Of the items that reflected male victimization on a general level, we selected the items with the highest component loadings. Then we selected items representing specific manifestations of male victimization, including different settings, perpetrators, and expressions. For each manifestation (e.g., male victimization in the media, male victimization by feminists), we again selected the items with the highest component loadings. We also selected the highest loading items illustrating zero-sum beliefs about gender discrimination. Lastly, we selected the reverse-coded items with the highest component loadings to provide within-scale attention checks. All analyses were conducted in R using the psych package [36].

## Results

The analysis of the eigenvalues and the scree plot (Fig 1) supported a one-dimensional structure of the proposed latent construct (i.e., BSS), with the first eigenvalue being considerably larger than the subsequent eigenvalues. As Horn's parallel analysis suggested that a second component might be evident, we examined a one-component and a two-component solution. However, while the first component explained 57% of variance, a second component explained merely 4.3% of additional variance. In addition, 67 items (89.3%) loaded highly on the first component (i.e., loadings > |.60|; [37,38]), indicating that one component was a good representation of the BSS construct. We then reduced the list of 75 items to 28 items. We selected 3 items reflecting male victimization on a general level, 22 items describing specific manifestations of anti-male discrimination, and 3 items reflecting zero-sum beliefs about gender discrimination. Six of the 28 selected items were reverse-coded. See S1 Table for the component loadings of the 75 items included in the Pilot Study.

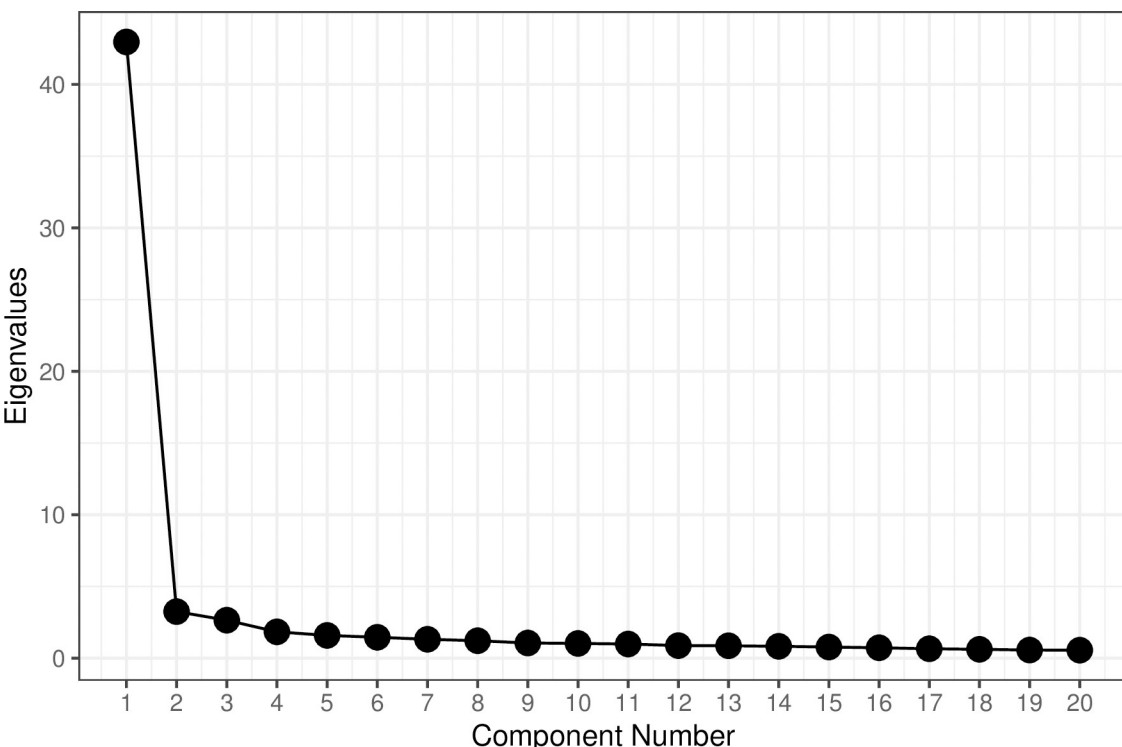

**Fig 1. Scree plot of the eigenvalues (Pilot Study).** Note: The number of components shown here was limited to 20. The real maximum number of components was 75.

## Study 1

In Study 1, we assessed the psychometric properties of the reduced subset of 28 items identified in the pilot study and selected a subset of 15 items that represented the BSS construct. We then examined the plausibility of a one-dimensional structure of the final set of items.

### Participants and procedure

We analyzed the responses of two hundred and nineteen participants (see Table 2 for sample characteristics). To determine sample size, we conducted power analyses based on the results from the pilot study. Specifically, we selected the item with the lowest factor loading, the item with the highest factor loading, and one item with an average factor loading. We then estimated the power that each of these items had achieved in the pilot study. The selected items had high test power (>.80) despite the small sample size. We thus felt confident that a similarly sized sample (n $\geq$ 112) would be sufficiently well-powered. We initially recruited 120 individuals for Study 1. However, after making slight changes to the wording of one item, we recruited 120 additional participants to ensure that rewording did not change the perceived meaning of the item. Given that the means of the reworded item were not significantly different across the two samples, we decided to collapse the two samples for subsequent analyses, $t(217) = -1.72$, $p = .09$.

### Data analysis

We used a maximum likelihood EFA to examine the psychometric properties of the 28 items selected using pilot study results. We computed the eigenvalues of the 28 by 28 item matrix and again determined the number of factors by examining the scree plot of the eigenvalues and conducting a parallel analysis. Then, we performed EFA on a one-factor and a two-factor solution. In the two-factor EFA, we used Promax to rotate the solution (an oblique rotation that allows factors to be correlated). EFA was performed in R using the fa function in the psych package [36].

Fifteen of the 28 items were selected to represent the BSS scale based on conceptual and statistical considerations. Specifically, we chose the items with the highest factor loadings within three conceptual domains, male victimization on a general level, specific manifestations of male victimization (e.g., specific contexts, perpetrators, and expressions), and zero-sum beliefs about gender discrimination. Finally, we selected the reverse-coded items with the highest factor loadings.

As a preliminary analysis of the psychometric structure of the final sets of items, we performed maximum likelihood CFA on a one-factor model. Model fit was assessed through chi-square, root-mean-square error of approximation (RMSEA), comparative fit index (CFI), Tucker-Lewis index (TLI), and standardized root mean square residuals (SRMR). We expected that a well-fitting model would meet most of the following criteria: RMSEA < .06, CFI > .95, TLI > .95, and SRMR < .08 [39]. CFA was performed in R using the lavaan package for latent variable analyses [40].

### Results

Consistent with the results from the pilot study, the scree plot suggested a one-dimensional solution (Fig 2). Also in line with the pilot study, parallel analysis suggested that a second or third factor might be evident. To explore that possibility, we examined both the one- and two-factor EFA solutions. The one-factor solution explained 58%, and the two-factor solution explained merely 2% of additional variance. For the one-factor solution, the fit indices were, *RMSR* = .05, *TLI* = .89, *RMSEA* = .09, 90% CI [.08, .09], whereas for the two-factor solution they were *RMSR* = .04, *TLI* = .92, *RMSEA* = .08, 90% CI [.06, .08]. The LR chi square for the

one-factor model was $LR(350) = 867.33$ and for the two-factor model it was $LR(323) = 686.51$. The difference between them was $LR(72) = 180.82$, $p < .0001$.

All item loadings from the one factor solution were in the expected direction, ranging in magnitude from .91 to .33, with a median loading of .79. The loadings in the Promax-rotated two-factor solution were also in the expected direction, with seventeen items loading highest on a factor reflecting male victimization on a general level and ten items loading on a factor marked by zero-sum beliefs about gender discrimination. A final item was weakly related to both factors. However, the two Promax-rotated factors correlated strongly in a positive direction, $r = .82$, supporting the notion that these two specific dimensions are aspects of the more general BSS dimension.

We therefore selected the final set of 15 items based on the one-factor solution. Three items described male victimization on a general level, ten items reflected specific manifestations of male victimization, and two items reflected zero-sum beliefs about gender discrimination. Three of these items were reverse-coded.

As expected, the dominant one-dimensional structure of the BSS scale was again found with the 15 items selected for the final BSS scale. The fit indices for the one factor solution suggested good fit, $CFI = .95$, $TLI = .95$, $RMSEA = .08$, 90% CI [.07, .10], $SRMR = .04$. The internal consistency of the BSS scale was very high, $Cronbach\ \alpha = .96$, 95% CI [.95, .97]. See S1 Table for the factor loadings of all 28 items included in Study 1.

## Study 2

In study 2, we sought to assess the factor structure of the final 15-item BSS scale with a different sample. We also began to test the conceptual and statistical validity of the BSS construct.

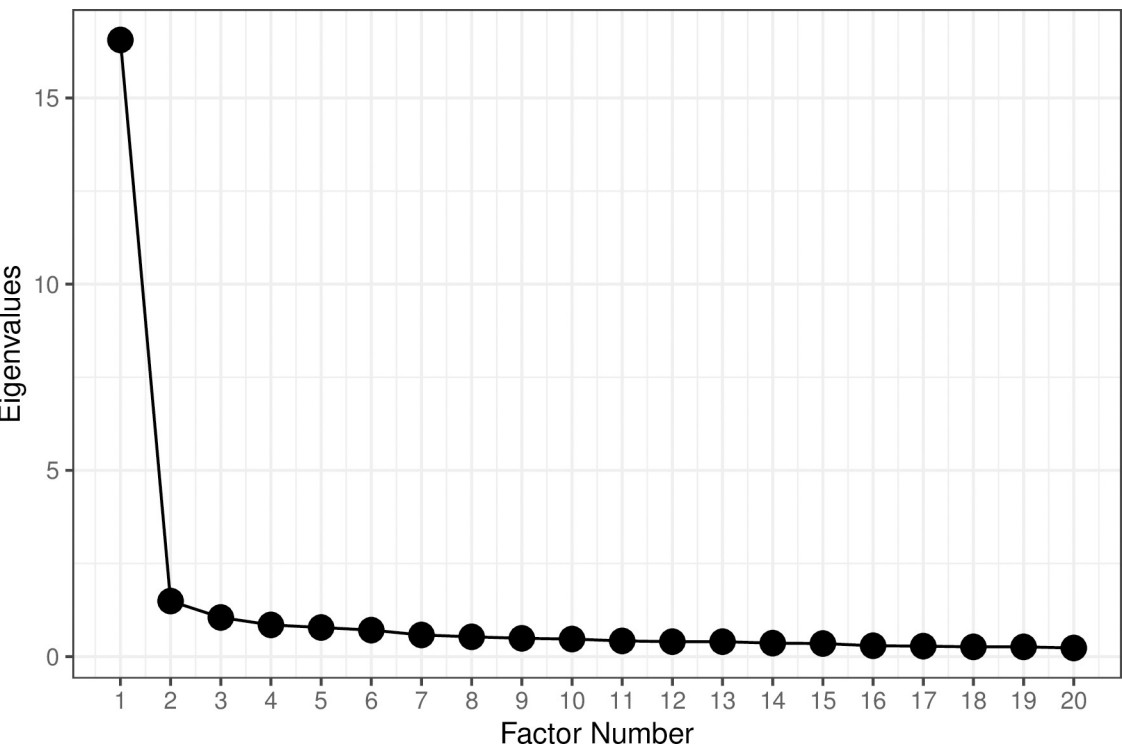

**Fig 2. Scree plot of the eigenvalues (Study 1).** Note: The number of factors shown here was limited to 20. The real maximum number of factors was 28.

We proposed that BSS is a subtle form of contemporary anti-female sexism that, unlike previous forms of sexism, masks negative attitudes towards women through a narrative of male victimhood. Accordingly, we expected BSS to be related to, yet distinct from, other forms of sexism. To support convergent validity, we examined the descriptive statistics (e.g., means, standard deviations) and expected that BSS would show a pattern of gender differences that is typical for contemporary sexism. Namely, men should have significantly higher BSS mean scores than women [41,42]. In addition, we examined the associations between BSS, traditional sexism, modern sexism, hostile sexism, and benevolent sexism. As a form of subtle, contemporary sexism, BSS should show strong associations with modern sexism and only weak to moderate associations with traditional sexism. BSS is also thought to conceal negative attitudes towards women. Thus, we expected strong associations of BSS with hostile sexism and weak associations with benevolent sexism.

We also began to assess discriminant validity by testing whether in factor analyses, BSS emerged as a distinct construct when analyzed alongside other forms of sexism. Further, if BSS is a subtle form of sexism, as we have proposed, it should not be readily recognized as such. Therefore, we expected the BSS scale to show less susceptibility to social desirability considerations than other forms of sexism.

Finally, to examine the test-retest reliability of the BSS, we contacted participants to complete the BSS scale again after two weeks.

## Participants and procedure

The sample for this study included three hundred and thirty-five participants (see Table 2 for the full sample characteristics). In addition to completing the BSS scale, participants were randomly assigned to complete two of the three most widely used sexism scales [41,43]. The sexism scales were presented on separate pages and randomized in order. All participants were asked to complete a social desirability measure [44].

The sample size was determined with consideration to sufficient numbers of participants for the correlations of the BSS scale with other sexism scales. We assumed high internal consistency of the BSS scale and each of the other sexism scales (Cronbach $\alpha < .80$), and we expected strong correlations ($r < |.60|$) between the BSS scale and contemporary forms of sexism (i.e., modern and hostile sexism). Based on these expectations, we determined that a sample size of at least 200 participation per correlation of the BSS scale with any other scale was sufficient [45].

To examine test-retest reliability, all participants were contacted two weeks after their initial participation. The responses of two hundred and six individuals who completed the BSS scale a second time were analyzed (see Table 2 for sample characteristics).

## Materials

In addition to the BSS scale, research material included three sexism measures: the 25-item Attitudes towards Women Scale to measure traditional sexism ($\alpha = .94$; [27]), the 8-item Modern Sexism Scale ($\alpha = .94$; [28]). and the 22-item Ambivalent Sexism Inventory, including the 11-item Hostile Sexism Scale ($\alpha = .95$) and the 11-item Benevolent Sexism Scale ($\alpha = .91$; [29]). For consistency, all scales were measured on 7-point scales (1 = *I strongly disagree*, 7 = *I strongly agree*). Higher scores indicated higher levels of sexism.

Social desirability was assessed using the 10-item short form of the Marlow-Crowne Social Desirability Scale ($\alpha = .72$; [44]). Participants were asked to give True-False answers to statements such as "I have never intensely disliked anyone". Scores could range from 0 to 10, with higher scores indicating higher levels of socially desirable response behavior.

## Data analysis

**Confirmatory factor analyses.** After inspecting item means and standard deviations by gender, we performed maximum likelihood CFA on a one-factor model to replicate the psychometric structure of the BSS scale in this new sample. Model fit was assessed through the criteria described in Study 1.

**Convergent validity.** We analyzed the descriptive statistics and examined gender differences in BSS mean scores. Then, we estimated Pearson product-moment correlations of the BSS scale with traditional, modern, hostile, and benevolent sexism. Based on previous studies examining the correlations of sexism scales [29,43], we determined that strong associations would be indicated by correlations > .60, moderate associations by correlations between .30 and .60, and weak associations by correlations < .30. To systematically assess the strength of the respective correlation coefficients and test whether BSS correlated more strongly with modern sexism than traditional sexism, and with hostile than benevolent sexism, we conducted coefficient comparisons [46,47]. Coefficient comparison analyses were performed in R using the "cocor" package [48].

**Discriminant validity.** To assess whether the psychometric structure of the BSS scale was related, but statistically distinct from other sexism measures, we performed a series of maximum likelihood CFAs. Each CFA included the BSS scale and one other sexism measure. In each analysis, we compared a two-factor model (i.e., a model where the BSS scale and the other sexism measure were correlated, but distinct factors), to a one-factor model (i.e., a model where the BSS scale and the other sexism measure converged on one common factor). We expected that the two-factor models would show better fit than the one-factor models, indicating that BSS was statistically distinct from the other scales. CFA model fit criteria [39] and Chi-square difference tests [49] were computed for model comparisons.

In addition, we calculated Pearson product-moment correlations between the social desirability scale and each sexism scale and compared these correlations. These analyses were performed separately for women and men as there is conflicting evidence about gender differences in the ability to recognize sexism as such [41,50].

**Test-retest reliability.** To investigate the test-retest reliability of the BSS scale, we calculated Pearson product-moment correlations between participants' mean scores at time one and time two.

## Results

**Confirmatory factor analyses.** The maximum likelihood CFA on a one-factor model yielded acceptable model fit: *CFI* = .93, *TLI* = .92, *RMSEA* = .10, 90% CI [.09, .11], *SRMR* = .04. However, these fit indices were below the model fit criteria that we had initially defined. We examined the possibility that a secondary factor might be detracting from the fit and found in a two-factor EFA that the three negatively worded items were more correlated with each other than with the positively worded items. We therefore repeated the CFA with a bi-factor model that accounted for item keying effects that were uncorrelated with the general BSS factor. This model yielded very good model fit, with most of the fit indices conforming to the defined fit criteria, *CFI* = .97, *TLI* = .95, *RMSEA* = .08, 90% CI [.07, .09], *SRMR* = .03. This indicates that the lack of fit of the one-factor model was based on response bias associated with the reverse-coded items, and not on a different theoretical structure of the BSS construct [51]. The internal consistency of the BSS scale was, again, very high, *Cronbach α* = .96, 95% CI [.96, .97].

In sum, these results verified the one-dimensional psychometric structure of the BSS scale. Table 3 summarizes item means and standard deviations by gender. Table 4 summarizes the standardized component and factor loadings of the final 15 items for each study (Pilot Study,

Study 1, Study 2). The final BSS scale, including instructions for its administration, is presented in Appendix A.

**Convergent validity.** The BSS scale's mean scores–like the scores of the other sexism measures–were significantly higher among men ($M = 3.45$, $SD = 1.50$) than among women ($M = 2.57$, $SD = 1.52$; $t(333) = -5.30$, $p < .0001$). Although the BSS scale correlated strongly with both modern sexism, $r(219) = .79$, 95% CI [.73, .83], $p < .0001$, and traditional sexism, $r(223) = .69$, 95% CI [.61, .75], $p < .0001$, coefficient comparison analyses confirmed that the association between BSS and modern sexism was significantly stronger than the association between BSS and traditional sexism, $z = 2.20$, $p = .028$. Also in line with our expectations, the BSS scale correlated more strongly with hostile sexism, $r(222) = .87$, 95% CI [.84, .90], $p < .0001$, than with benevolent sexism, $r(222) = .43$, 95% CI [.31, 53], $p < .0001$; coefficient comparison confirmed that these correlations were significantly different, $z = 10.80$, $p < .0001$. See Fig 3 for the distribution of BSS scores by gender and Table 5 for the means, standard deviations, and correlations of all sexism scales.

**Discriminant validity.** CFAs including the BSS scale and each of the other sexism measures provided support for the discriminant validity of the BSS scale. The model fit criteria and the chi-square difference tests consistently supported two-factor models over one-factor models, indicating that BSS is related to, but distinct from other forms of sexism. See Table 6 for all model comparisons.

Social desirability was moderate among women ($M = 3.84$, $SD = 2.47$) and men ($M = 3.78$, $SD = 2.55$), $t(333) = 0.21$, $p = .836$. An analysis of the correlations between each of the sexism measures and social desirability further supported the discriminant validity of BSS. As expected, the BBS scale did not correlate significantly with social desirability among women, $r(143) = .07$, 95% CI [-.10, .23], $p = .440$, or among men, $r(186) = -.10$, 95% CI [-.23, .05], $p = .193$. In contrast, all other sexism scales showed correlations with social desirability, either among women or among men (Table 7; note that among men, one outlier was excluded from the correlation between hostile sexism and social desirability).

**Test-retest reliability.** Finally, BSS scores at time one correlated strongly with mean scores at time two, $r(203) = .92$, 95% CI [.89, .94], providing support for the test-retest reliability of the BSS scale.

**Table 3. BSS scale item means and standard deviations by gender.**

|  | Women | | Men | |
| --- | --- | --- | --- | --- |
|  | M | SD | M | SD |
| In the US, discrimination against men is on the rise. | 2.56 | 1.90 | 3.69 | 1.88 |
| Men are not particularly discriminated against.* | 2.68 | 1.81 | 3.59 | 1.85 |
| If anything, men are more discriminated against than women these days. | 2.17 | 1.67 | 2.93 | 1.88 |
| Giving women more rights to women often requires taking away men's rights. | 2.17 | 1.64 | 2.68 | 1.77 |
| Under the guise of equality for women, men are actually being discriminated against. | 2.47 | 1.85 | 3.40 | 1.90 |
| In the pursuit of women's rights, the government has neglected men's rights. | 2.51 | 1.89 | 3.26 | 1.90 |
| Nowadays, men don't have the same chances in the job market as women. | 2.13 | 1.64 | 2.74 | 1.73 |
| Feminism is about favoring women over men. | 2.57 | 1.95 | 3.66 | 2.14 |
| Feminism does not discriminate against men.* | 2.92 | 2.96 | 4.15 | 2.00 |
| All in all, men have more responsibilities and fewer benefits. | 2.47 | 1.81 | 3.42 | 1.86 |
| In today's society, women can say things that men are not allowed to say. | 3.35 | 2.10 | 3.96 | 1.97 |
| It is evident that the media is biased against men. | 2.57 | 1.88 | 3.47 | 1.95 |
| In today's society, men are often punished for acting manly. | 2.54 | 1.89 | 3.63 | 1.99 |
| All in all, men are well respected in today's society.* | 2.35 | 1.40 | 2.96 | 1.63 |
| While women can use the "gender-card" to get ahead, men can't. | 3.08 | 2.17 | 4.24 | 2.08 |

**Table 4. Standardized component and factor loadings of the final BSS scale's items by study.**

| | Pilot (N = 112) | Study 1 (N = 220) | Study 2 (N = 335) |
|---|---|---|---|
| In the US, discrimination against men is on the rise. | .90 | .86 | .87 |
| Men are not particularly discriminated against.* | -.81 | -.55 | -.68 |
| If anything, men are more discriminated against than women these days. | .88 | .86 | .91 |
| Giving women more rights often requires taking away men's rights. | .86 | .79 | 75 |
| Under the guise of equality for women, men are actually being discriminated against. | .88 | .91 | .92 |
| In the pursuit of women's rights, the government has neglected men's rights. | .91 | .89 | .87 |
| Nowadays, men don't have the same chances in the job market as women. | .80 | .78 | .77 |
| Feminism is about favoring women over men. | .80 | .78 | .83 |
| Feminism does not discriminate against men.* | -.76 | -.66 | -.74 |
| All in all, men have more responsibilities and fewer benefits. | .84 | .81 | .83 |
| In today's society, women can say things that men are not allowed to say. | .78 | .75 | .68 |
| It is evident that the media is biased against men. | .92 | .84 | .87 |
| In today's society, men are often punished for acting manly. | .80 | .81 | .83 |
| All in all, men are well respected in today's society.* | -.69 | -.59 | -.61 |
| While women can use the "gender-card" to get ahead, men can't. | .73 | .79 | .79 |

Note. * signifies reverse-coded items.

## Study 3

Given the content of the BSS scale and gender differences in mean BSS scores, as a final step, we sought to assess whether the underlying construct had similar meaning for women and

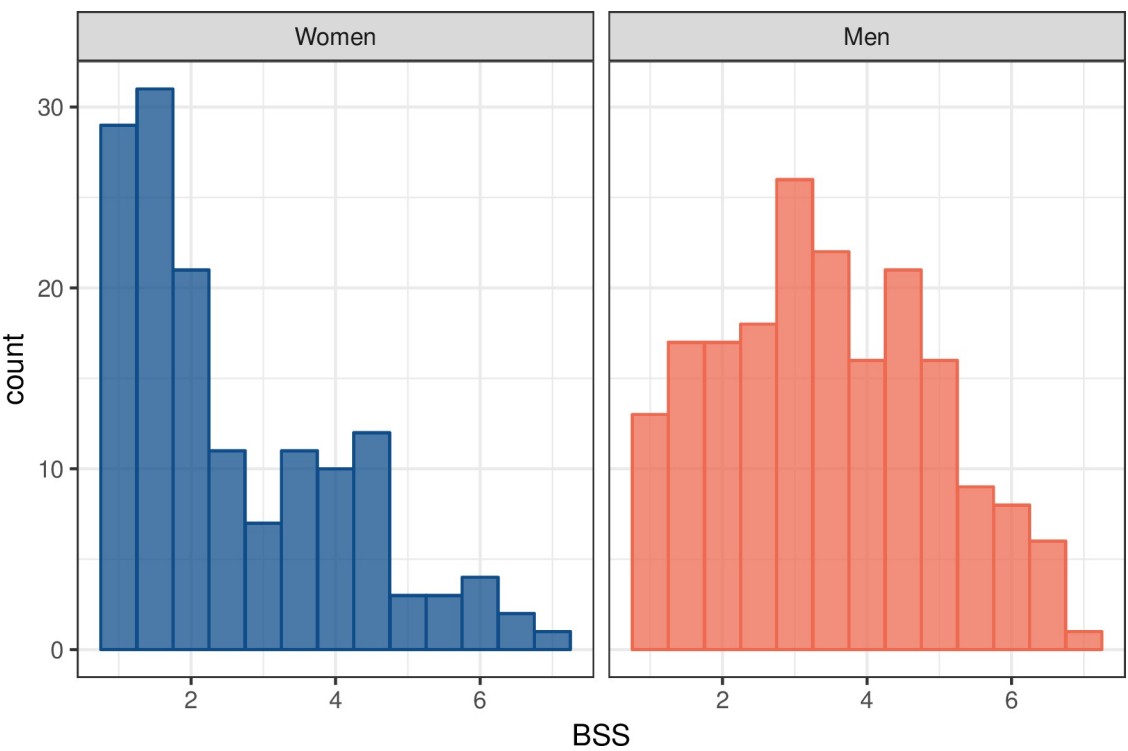

**Fig 3. Distribution of the BSS scores by gender.**

**Table 5. BSS scale means, standard deviations, and correlations with other sexism scales (traditional sexism, modern sexism, hostile sexism, and benevolent sexism).**

| | Women | Men | | | | | | | |
|---|---|---|---|---|---|---|---|---|---|
| | M (SD) | M (SD) | t | p | BSS | TS | MS | HS | BS |
| Belief in sexism shift (BSS) | 2.57 (1.52) | 3.45 (1.50) | -5.30 | < .0001 | 1 | | | | |
| Traditional sexism (TS) | 2.01 (1.04) | 2.27 (1.01) | -1.82 | .071 | .69 | 1 | | | |
| Modern sexism (MS) | 2.51 (1.37) | 3.41 (1.46) | -4.72 | < .0001 | .79 | .61 | 1 | | |
| Hostile sexism (HS) | 2.58 (1.41) | 3.32 (1.52) | -3.75 | < .0001 | .87 | .74 | .75 | 1 | |
| Benevolent sexism (BS) | 3.09 (1.47) | 3.61 (1.33) | -2.78 | .006 | .43 | .58 | .41 | .49 | 1 |

men. Analysis of measurement invariance (AMI) is an important, yet often neglected aspect in the development and validation of new measurement scales. Measurement invariance across groups allows researchers to conclude that mean differences in group scores reflect differences in the latent construct, and are not the result of a third, unknown factor [52,53].

## Participants

To obtain a large sample for analysis, we collapsed the samples from the previous studies (Pilot Study, Study 1, and Study 2). The collapsed sample included 664 individuals (see Table 2 for the sample characteristics).

## Data analysis

We performed AMI within a structural equation modelling framework using the R "sem-Tools" package [54]. AMI is a stepwise process and requires a sequence of model tests where each model is tested against its predecessor. The first step is the assessment of configural invariance, which indicates equivalence in model form across groups (e.g., similar number of factors, similar pattern in free and fixed loadings). The second step is the assessment of metric invariance, which indicates equivalence in factor loadings across groups. The third step is the assessment of scalar invariance, which indicates equivalence of item intercepts across groups. We examined configural, metric, and scalar measurement invariance of the BSS scale for women and men. First, we computed a maximum likelihood one-

**Table 6. Discriminant validity of the BSS scale compared to other sexism scales (traditional sexism, modern sexism, hostile sexism, and benevolent sexism).**

| | X² | df | ΔX² | Δdf | CFI | TLI | RMSEA |
|---|---|---|---|---|---|---|---|
| **BSS and traditional sexism** | | | | | | | |
| Discriminant two-factor model | 1656.30 | 739 | | | .86 | .85 | .07 |
| Unitary one-factor model | 2603.64 | 740 | 947.34*** | 1 | .72 | .70 | .11 |
| **BSS and modern sexism** | | | | | | | |
| Discriminant two-factor model | 721.01 | 229 | | | .90 | .89 | .10 |
| Unitary one-factor model | 1124.73 | 230 | 403.72*** | 1 | .82 | .80 | .13 |
| **BSS and hostile sexism** | | | | | | | |
| Discriminant two-factor model | 976.34 | 298 | | | .89 | .88 | .10 |
| Unitary one-factor model | 1238.08 | 299 | 261.74*** | 1 | .84 | .83 | .12 |
| **BSS and benevolent sexism** | | | | | | | |
| Discriminant two-factor model | 1076.12 | 298 | | | .84 | .83 | .11 |
| Unitary one-factor model | 1998.77 | 299 | 922.65*** | 1 | .66 | .63 | .16 |

*** $p < .0001$.

**Table 7. Pearson product moment correlations of the sexism scales with social desirability by gender.**

|  | Social desirability | |
|---|---|---|
|  | **Women** | **Men** |
| **Belief in Sexism Shift** | .07 | -.10 |
| **Traditional Sexism** | .28 | .12 |
| **Modern Sexism** | .18 | .08 |
| **Hostile Sexism** | .10 | -.22* |
| **Benevolent Sexism** | .24* | -.05 |

*p < .05.

factor CFA on the full sample to gain estimates for the baseline model fit criteria. Then, we compared a sequence of three invariance models. The fundamental structure of all models was the one-factor solution. In Model 1, we assessed configural invariance, which was indicated by model fit criteria similar to the baseline criteria, *CFI* = .95, and *RMSEA* = .08. In Model 2, we constrained the factor loadings to be equivalent for women and men. Then, we compared the fit of Model 1 and Model 2. The criteria to assume metric invariance are a non-significant change in Chi-square, $\Delta X^2$, $\Delta CFI > $ -.02, and $\Delta RMSEA < .03$ [55]. In Model 3, we constrained the item intercepts to be equivalent for women and men and compared the fit of Model 2 and Model 3. Criteria for scalar invariance are a non-significant change in Chi square, $\Delta X^2$, $\Delta CFI > $ -.01, and $\Delta RMSEA < .01$ [55].

## Results

Results indicated measurement invariance of the BSS scale across participant gender. Model 1 supported configural invariance across gender, $X^2(180) = 631.27$, *CFI* = .95, and *RMSEA* = .09. The comparison of Model 1 and Model 2 supported the notion of metric invariance; the chi-square values were not statistically different, $\Delta X^2(14) = 17.53$, *p* = .229. The observed changes $\Delta CFI < .0001$ and $\Delta RMSEA = .003$ were consistent with the criteria for metric invariance described above. The comparison of Model 2 and Model 3 supported the notion of scalar invariance. The chi-square values were not statistically different, $\Delta X^2(14) = 13.76$, *p* = .468; changes in $\Delta CFI < .0001$ and $\Delta RMSEA = .003$ were in line with the criteria for scalar invariance described above (AMI performed with a bi-factor model that accounted for item keying effects produced similar results).

We can thus conclude that BSS has similar meaning for women and men and that mean gender differences in BSS can be interpreted as mean differences in the latent construct. See Table 8 for the results of AMI.

## Discussion

In this paper, we defined belief in sexism shift as a new manifestation of anti-female sexism and introduced a scale to measure it. Consistent with our definition, BSS emerged as a unidimensional construct including assertions about the current victimization of men, the multitude of ways in which male victimization is manifested, and the belief that male victimization is the result of women's recent advancement.

Our findings further indicate that the BSS scale measures a new, distinct form of sexism. Past definitions of sexism have been built around people's attitudes towards women and/or their opinions on the state of discrimination against women. In contrast, BSS focuses on men and male victimhood. In line with our predictions, BSS consistently emerged as a distinct

**Table 8. Analyses of measurement invariance of the BSS scale across gender.**

|  | Model 1 Configural invariance | Model 2 Metric invariance | Model 3 Scalar invariance |
|---|---|---|---|
| **Chi square (df)** | 631.27 (180) | 648.80 (194) | 662.55 (208) |
| **CFI** | .95 | .94 | .94 |
| **RMSEA** | .09 | .08 | .08 |
| **Model comparison** | - | Model 1 | Model 2 |
| **Δ Chi square (Δ df)** | - | 17.53 (14) | 13.76 (14) |
| **P** |  | .229 | .468 |
| **ΔCFI** | - | > -.02 | > -.01 |
| **ΔRMSEA** | - | < .03 | < .01 |
| **Decision** | - | Accept | Accept |

construct in our analyses. In addition, our results demonstrated that the BSS scale was a stable and reliable measure of BSS across different samples, time, and participant gender.

## BSS as contemporary sexism

Supporting our claim that BSS is a new form of sexism, gender differences in BSS mean scores were similar to those of other sexism measures, with men showing higher mean levels of BSS than women. Moreover, the BSS scale was positively associated with previous anti-female sexism measures, albeit to different extents. In line with the idea that BSS constitutes a type of subtle, contemporary sexism, the BSS scale was more strongly correlated with modern than traditional sexism. Also supporting our notion that BSS upholds negative attitudes towards women, there was a stronger association with hostile than benevolent sexism.

## BSS as subtle sexism

Endorsing sexist beliefs is generally considered reprehensible [41]. As such, sexism measures are often susceptible to social desirability. Compared to previous sexism measures, the BSS scale was less sensitive to social desirability concerns. Given its focus on the victimization of men and the acknowledgment of ongoing gender discrimination, it is possible that BSS is not readily seen as anti-female sexism. Thus, BSS may be a particularly subtle form of sexism, a characteristic that may make the BSS scale a uniquely useful tool for studying sexism and its downstream consequences in the current social landscape.

## Future directions

The BSS scale is the first to concisely capture an attitude that appears to have gained momentum in recent years [7,11,34]. As such, it can be used to advance our understanding of contemporary sexism and help explain emerging gender issues. This new scale can help shed light on whether the prevalence of BSS is growing in the general population, as survey data have suggested [11,56], as well as to identify important social, cultural, and psychological phenomena associated with BSS.

Like other forms of sexism, BSS may reflect relatively stable differences between individuals. In future research, the BSS scale can be used to map the consequences of this individual difference on a variety of outcomes. However, BSS may also be the result of recent societal changes regarding the status of women and men. Thus, future research can also use the BSS scale to examine the antecedents of this belief.

**Potential consequences of BSS.** Sexism has been associated with a wide array of negative consequences that disproportionally affect women [57,58]. Like other forms of anti-female

sexism, BSS should also predict negative outcomes for women. Previous research has shown that priming men with anti-male bias increases discrimination against female candidates for a job [15]. We expect that higher BSS endorsement should have a similar effect, leading to more negative evaluations of women than men in the workplace and beyond. However, the BSS scale may be more useful than previous sexism measures in predicting not only negative outcomes for women, but also an active motivation to obtain positive outcomes for men. Future research could use the BSS scale to explore the role of BSS in people's opposition to diversity measures when they help women, but their support for such measures when they favor men [59].

The BSS scale can also be used to examine whether BSS has had an influence, over and above other forms of sexism (e.g., hostile sexism; [60]), on current ideological shifts and the rise of extremist beliefs across the globe. For example, the portrayal of historically privileged groups as victims, an idea that is consistent with BSS, has been associated with support for populist Right-wing parties [61,62].

**Potential causes of BSS.** The BSS scale will be a valuable tool in determining whether and how different gender-relevant contextual factors have contributed to the emergence and endorsement of BSS. Perceptions of male victimization may stem, in part, from recent changes regarding the status of women and men [21]. For example, women's greater academic achievement [63] and steady influx into the labor force [64] may be perceived as costing men educational opportunities and jobs. Likewise, diversity initiatives and public policy aimed at improving conditions for women–but not men (e.g., women quotas)–may be seen as disadvantaging men.

The BSS scale can also be used to explore the extent to which self-enhancement motivations are associated with higher BSS endorsement among men. Past research has shown, for example, that claiming victimhood may protect self-esteem by allowing people to attribute negative outcomes (e.g., unemployment, poor performance) to unfair external conditions or discrimination [65]. Moreover, BSS may be a consequence of competitive victimhood, that is, men may be more likely to see themselves as victims of discrimination when confronted with continuing discrimination against women [66,67].

Importantly, BSS is likely part of a more pervasive phenomenon where members of historically advantaged groups come to see themselves as the new victims of discrimination. For example, the advancement of Black Americans has recently led to perceptions of anti-White bias and "reverse racism" [68,69]. Paralleling beliefs about changes in gender discrimination over time [10], studies in the U.S. show that Whites see racism against Black Americans as decreasing and racism against White Americans as increasing [70]. This suggests that, as with gender, there may be a belief in racism shift, and the BSS scale may serve as a model for similar measures of contemporary racism.

## Limitations

Although our samples were relatively small, statistical simulation studies suggest that the sample sizes were appropriate given the concise nature of our construct and data (e.g., one-dimensional factor structure, high factor-to-items-ratio, high communalities; [37,38]). However, our samples consisted mostly of White/ European Americans and the relatively small numbers of Asian, Black/African, and Hispanic/Latin Americans did not allow us to assess whether the BSS construct was similar for different ethnic groups. Previous research has indicated that sexism measures may have different meanings for ethnic minorities [43]. Thus, future research should analyze the reliability and validity of the BSS scale for different ethnic and racial groups, both within and outside of the U.S.

Despite a clear theoretical and psychometric distinction between the BSS scale and previous sexism measures, the strong correlation between this new scale and the hostile sexism scale

should be addressed. It is possible that those who endorse BSS blame women for the hardships of today's men. Likewise, hostile sexists may be more likely to view the world as a place that systematically victimizes men. Whether and when BSS leads to hostility towards women or hostility toward women leads to BSS should be examined in future research.

In addition to the strong association between the BSS scale and hostile sexism, the correlations between previous sexism measures were also stronger in our data than what has previously been reported [29,43]. It is possible that our design contributed to an overall overestimation of the correlations between different sexism scales, including the BSS scale. Although all sexism measures were counterbalanced and presented on separate questionnaire pages, they were presented sequentially. This may have led to similar response patterns across scales and, consequently, stronger correlations. Future research should continue testing the associations between the BSS scale and other forms of sexism, including, but not limited to the ones examined in the present research.

## Conclusion

In the present research, we introduced and defined the *belief in sexism shift* (BSS) as a new, contemporary form of sexism and developed a scale to measure it: the BSS scale. As expected, the BSS scale was related to, yet distinct from, traditional sexism, modern sexism, and ambivalent sexism. It also was stable across samples, time, and participant gender. In addition, the BSS scale was less susceptible to social desirability concerns than other sexism measures, suggesting that BSS may be a particularly subtle form of anti-female sexism. As the first concise, reliable measure of BSS, the BSS scale will be a valuable tool to enhance our understanding of an emerging and potentially growing belief that may have unprecedented consequences for women.

## Appendix A. The Belief in sexism shift scale

| **How much do you agree with the following statements?** |
| --- |
| There are no right or wrong answers. We are interested in *your* opinion! |
| 1 = *I strongly disagree*, 7 = *I strongly agree* |
| In the US, discrimination against men is on the rise. |
| Men are not particularly discriminated against.* |
| If anything, men are more discriminated against than women these days. |
| Giving women more rights often requires taking away men's rights. |
| Under the guise of equality for women, men are actually being discriminated against. |
| In the pursuit of women's rights, the government has neglected men's rights. |
| Nowadays, men don't have the same chances in the job market as women. |
| Feminism is about favoring women over men. |
| Feminism does not discriminate against men.* |
| All in all, men have more responsibilities and fewer benefits. |
| In today's society, women can say things that men are not allowed to say. |
| It is evident that the media is biased against men. |
| In today's society, men are often punished for acting manly. |
| All in all, men are well respected in today's society.* |
| While women can use the "gender-card" to get ahead, men can't. |

Note 1. * signifies reverse-coded items.
Note2. We recommend administering the items in a randomized order.

## Supporting information

**S1 Table. Standardized component and factor loadings of all items by study.**
(DOCX)

**S2 Table A Study material for the pilot study B Study material for Study 1 C Study material for Study 2.**
(DOCX)

## Author Contributions

**Conceptualization:** Miriam K. Zehnter, Francesca Manzi, Patrick E. Shrout, Madeline E. Heilman.

**Data curation:** Miriam K. Zehnter.

**Formal analysis:** Miriam K. Zehnter, Patrick E. Shrout.

**Funding acquisition:** Miriam K. Zehnter.

**Investigation:** Miriam K. Zehnter.

**Methodology:** Miriam K. Zehnter, Francesca Manzi, Patrick E. Shrout, Madeline E. Heilman.

**Project administration:** Miriam K. Zehnter.

**Resources:** Miriam K. Zehnter, Francesca Manzi.

**Software:** Miriam K. Zehnter.

**Supervision:** Miriam K. Zehnter, Francesca Manzi, Patrick E. Shrout, Madeline E. Heilman.

**Validation:** Miriam K. Zehnter, Francesca Manzi, Patrick E. Shrout.

**Visualization:** Miriam K. Zehnter.

**Writing – original draft:** Miriam K. Zehnter, Francesca Manzi.

**Writing – review & editing:** Miriam K. Zehnter, Francesca Manzi, Patrick E. Shrout, Madeline E. Heilman.

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
