## [Decision Letter · Decision Letter 0]

7 Jan 2021

PONE-D-20-30781

Belief in Sexism Shift: Defining a new form of contemporary sexism and introducing the belief in sexism shift scale (BSS scale)

PLOS ONE

Dear Dr. Zehnter,

Thank you for submitting your manuscript to PLOS ONE. After careful consideration, we feel that it has merit but does not fully meet PLOS ONE’s publication criteria as it currently stands. Therefore, we invite you to submit a revised version of the manuscript that addresses the points raised during the review process.

Please make the paper clearer following reviewer #2 suggestions, moving significant part of theory in the introduction section and making a clearer description of the method.

We look forward to receiving your revised manuscript.

Kind regards,

Marta Panzeri, Ph.D.

Academic Editor

PLOS ONE

Journal Requirements:

"Review Board: NYU WSQ

Approval number: IRB-FY2017-783

Informed consent was obtained written by checking a box in the online survey.".   

Reviewers' comments:

Reviewer's Responses to Questions

**Comments to the Author**

1. Is the manuscript technically sound, and do the data support the conclusions?

Reviewer #1: Yes

Reviewer #2: Yes

2. Has the statistical analysis been performed appropriately and rigorously? 

Reviewer #1: Yes

Reviewer #2: Yes

3. Have the authors made all data underlying the findings in their manuscript fully available?

Reviewer #1: Yes

Reviewer #2: Yes

4. Is the manuscript presented in an intelligible fashion and written in standard English?

Reviewer #1: Yes

Reviewer #2: Yes

5. Review Comments to the Author

Reviewer #1: I would like to see this article published in this journal. It makes an original contribution to the psychosocial literature, offering an adequate and well scrutinized discussion about an important and current topic. The methodology is well described, the analyses have been conducted rigorously, and the discussion section is clearly discussed.

Reviewer #2: The paper presents a new form of sexism, called Belief in sexism shift (BSS), a form of sexism that is main connected to anti-female sexism. Authors provide the development of a new promising scale measuring the BSS which could be useful in the study of the new forms of subtle sexism in our society. The statistical plan and the analysis are very rigorous and well conducted in my opinion. Overall, an interesting manuscript with some new important insights.

My main concern is about the readability of the paper. The introduction about BSS is well written, but quite long, dispersive and hard to follow. At the beginning it feels like BSS is a legitimate and motivated behavior but then, through the introduction and the discussion, it seems that authors condamned BSS. I feel that authors should just describe this behaviors, the features and how it is different from other forms of sexism and why it is important, which they actually do, but in a very long and confusing way. Also some aspects of the discussion (potential causes of of BSS and potential consequences of BSS) should be moved in the Introduction section, because they feel very important, and are not really a consequence of the work results. Authors should explain their results in the light of these consideration, not vice versa.

Finally, the three studies are promising and really well conducted from a methodological point of view. But, it's really hard to follow the description of participants and data analysis for all the studies (they are reported like 4 times..). Authors should find another easier and clearer way to present the study, for example to present participants description only in table, and to create a statistical analysis plan section with the methods for each study presented divided in bulletpoints.

Besides these aspects, a very interesting and useful papers for gender studies scholars and researchers.

6. PLOS authors have the option to publish the peer review history of their article (what does this mean?). If published, this will include your full peer review and any attached files.

Reviewer #1: No

Reviewer #2: No

---

## [Author Response · Author response to Decision Letter 0]

19 Feb 2021

Editor’s comments:

Dear Dr. Zehnter,

Thank you for submitting your manuscript to PLOS ONE. After careful consideration, we feel that it has merit but does not fully meet PLOS ONE’s publication criteria as it currently stands. Therefore, we invite you to submit a revised version of the manuscript that addresses the points raised during the review process.

Please make the paper clearer following reviewer #2 suggestions, moving significant part of theory in the introduction section and making a clearer description of the method.

Authors’ response

Dear Dr. Panzeri,

Thank you for considering our manuscript “Belief in Sexism Shift: Defining a new form of contemporary sexism and introducing the belief in sexism shift scale (BSS scale)”. We are very happy to hear that the reviewers found the paper interesting and that they see our scale as an important contribution to the literature. Please find enclosed a new version of our manuscript, which we have carefully revised based on your comments and those of the reviewers (in particular, reviewer #2). As requested, we have also included a version of the manuscript with tracked changes. Finally, to ensure that our manuscript meets PLOS ONE’s style requirements, we have amended our ethics statement to include the full name of the ethics committee that approved our studies. 

Below we provide a response to each comment made by reviewers. 

Reviewer #1 comments:

I would like to see this article published in this journal. It makes an original contribution to the psychosocial literature, offering an adequate and well scrutinized discussion about an important and current topic. The methodology is well described, the analyses have been conducted rigorously, and the discussion section is clearly discussed.

Authors’ response

Thank you very much for endorsing our manuscript and supporting our work. We, too, believe that the belief in sexism shift is an important and timely topic of research. We hope that the BSS scale will be a valuable tool for more reliably investigating the causes and consequences of this new belief. 

Reviewer #2 comments:

The paper presents a new form of sexism, called Belief in sexism shift (BSS), a form of sexism that is main connected to anti-female sexism. Authors provide the development of a new promising scale measuring the BSS which could be useful in the study of the new forms of subtle sexism in our society. The statistical plan and the analysis are very rigorous and well conducted in my opinion. Overall, an interesting manuscript with some new important insights.

My main concern is about the readability of the paper. The introduction about BSS is well written, but quite long, dispersive and hard to follow. At the beginning it feels like BSS is a legitimate and motivated behavior but then, through the introduction and the discussion, it seems that authors condamned BSS. I feel that authors should just describe this behaviors, the features and how it is different from other forms of sexism and why it is important, which they actually do, but in a very long and confusing way. Also some aspects of the discussion (potential causes of of BSS and potential consequences of BSS) should be moved in the Introduction section, because they feel very important, and are not really a consequence of the work results. Authors should explain their results in the light of these consideration, not vice versa.

Finally, the three studies are promising and really well conducted from a methodological point of view. But, it's really hard to follow the description of participants and data analysis for all the studies (they are reported like 4 times..). Authors should find another easier and clearer way to present the study, for example to present participants description only in table, and to create a statistical analysis plan section with the methods for each study presented divided in bulletpoints.

Besides these aspects, a very interesting and useful papers for gender studies scholars and researchers.

Authors’ response

Thank you very much for your encouraging words and helpful feedback. As you will see in the revised manuscript, we have considered your comments carefully and made several important changes. Below, we describe the main revisions to our manuscript based on your suggestions:

1) We have now revised the entire manuscript for clarity and conciseness. We agree that the introduction was lengthy and, at times, repetitive. Thus, we have made an effort to shorten it (from 7 to 6 pages) and to ensure that the structure is more cohesive (avoiding unnecessary reiterations of ideas). We also have revised our language throughout to exclude words or phrases that could be perceived as condemning, rather than describing, the belief in sexism shift (BSS). 

2) We have carefully considered your suggestion to relocate parts of the discussion of potential causes and consequences of BSS to the introduction. While we agree that these aspects are important, we have decided not to move these sections for several reasons. First, in light of your suggestions, we have sought to be as clear as possible about the goals of this research in the introduction – that is, to define BSS and to develop a new scale to measure it. Consequently, we thought that describing potential causes and consequences of BSS in the introduction would not only add confusion, but it might also be misleading, as readers could expect that the research we present will also examine the causes and consequences of BSS (which it does not). Importantly, given that these ideas were not directly tested in our paper, the causes and consequences of BSS remain speculative. Indeed, in the discussion we mention several potential causes (e.g., self-enhancement motivation, competitive victimhood) and consequences (e.g., political extremity, increased discrimination of women) of BSS, but we do not yet know whether they are in fact related. We hope that the BSS scale will be a valuable tool in determining the extent to which the factors we describe (and/or other factors) are associated with BSS and, as such, we think that the most appropriate place to discuss these possibilities is in the “Future Directions” section of the discussion. Nevertheless, considering your comments and those of the editor, we have revised the manuscript to describe more explicitly how we think future research can use the BSS scale developed in this research to study the causes and consequences of this belief. We would like to point out that if, after reading our revised manuscript, you are not fully convinced of our reasoning, we are happy to reconsider this point. 

3) We have made important revisions to the methods section for clarity. In particular, we have updated Table 2 to also include the full information of each sample. As you will see, in-text descriptions of research participants have been reduced to a minimum. In addition, we have included a new table (Table 1) that provides an overview of the four studies included in this manuscript, their aims, and the statistical analyses conducted. 

We would like to thank the editor and reviewers for offering us the opportunity to revise our manuscript. We hope to have addressed your concerns in the revised version of this paper. We strongly believe your suggestions have significantly improved the quality and potential impact of our research. 

Kind regards,

Miriam Zehnter, Francesca Manzi, Patrick Shrout, and Madeline Heilman

---

## [Decision Letter · Decision Letter 1]

25 Feb 2021

Belief in Sexism Shift: Defining a new form of contemporary sexism and introducing the belief in sexism shift scale (BSS scale)

PONE-D-20-30781R1

Dear Dr. Zehnter,

We’re pleased to inform you that your manuscript has been judged scientifically suitable for publication and will be formally accepted for publication once it meets all outstanding technical requirements.

Kind regards,

Marta Panzeri, Ph.D.

Academic Editor

PLOS ONE

Additional Editor Comments (optional):

Reviewers' comments:

Reviewer's Responses to Questions

**Comments to the Author**

1. If the authors have adequately addressed your comments raised in a previous round of review and you feel that this manuscript is now acceptable for publication, you may indicate that here to bypass the “Comments to the Author” section, enter your conflict of interest statement in the “Confidential to Editor” section, and submit your "Accept" recommendation.

Reviewer #2: All comments have been addressed

2. Is the manuscript technically sound, and do the data support the conclusions?

Reviewer #2: Yes

3. Has the statistical analysis been performed appropriately and rigorously? 

Reviewer #2: Yes

4. Have the authors made all data underlying the findings in their manuscript fully available?

Reviewer #2: Yes

5. Is the manuscript presented in an intelligible fashion and written in standard English?

Reviewer #2: Yes

6. Review Comments to the Author

Reviewer #2: (No Response)

7. PLOS authors have the option to publish the peer review history of their article (what does this mean?). If published, this will include your full peer review and any attached files.

Reviewer #2: **Yes: **Lilybeth Fontanesi

---

## [Editor Report · Acceptance letter]

3 Mar 2021

PONE-D-20-30781R1 

Belief in Sexism Shift:Defining a new form of contemporary sexism and introducing the belief in sexism shift scale (BSS scale) 

Dear Dr. Zehnter:

I'm pleased to inform you that your manuscript has been deemed suitable for publication in PLOS ONE. Congratulations! Your manuscript is now with our production department. 

Kind regards, 

on behalf of

Dr. Marta Panzeri 

Academic Editor

PLOS ONE